Integrated bioinformatics analysis reveals dynamic candidate genes and signaling pathways involved in the progression and prognosis of diffuse large B-cell lymphoma

Charwudzi Alice
Meng Ye
Hu Linhui
Ding Chen
Pu Lianfang
Li Qian
Xu Mengling
Zhai Zhimin zzzm889@163.com
Xiong Shudao xshdao@ahmu.edu.cn
Department of Hematology/Hematological Lab, The Second Hospital of Anhui Medical University , Hefei, Anhui , China
Ge Jianye
Electronic publication date: 2021 Nov 2
Publication date: 2021
Volume: 9
Electronic Location ID: e12394
Received 2021 Jun 28; Accepted 2021 Oct 5
Copyright: © 2021 Charwudzi et al.
Copyright year: 2021
Copyright holder: Charwudzi et al.
License: This is an open access article distributed under the terms of the Creative Commons Attribution License, which permits unrestricted use, distribution, reproduction and adaptation in any medium and for any purpose provided that it is properly attributed. For attribution, the original author(s), title, publication source (PeerJ) and either DOI or URL of the article must be cited.
License URL: https://creativecommons.org/licenses/by/4.0/

Keywords: Diffuse large B-cell lymphoma, Integrated bioinformatic analysis, Hub genes, Ribosome, COVID-19, Immune cells

Funding: Key Research and Development Plan of Anhui Province, China 201904a07020058 National Science Foundation of China 81272259, 81670179 Major Subject of Science and Technology of Anhui Province, China 201903a07020030 Higher School of Anhui Provincial Natural Science Research Project, China KJ2018A0198 The Foundation of Anhui Medical University, China 2019xkj134 Basic and Clinical Cooperative Research Promotion Plan of Anhui Medical University, China 2020xkjT021 This work was supported by grants from the Key Research and Development Plan of Anhui Province, China (grant number 201904a07020058); National Science Foundation of China (81272259, 81670179); Major Subject of Science and Technology of Anhui Province, China (grant number 201903a07020030); Higher School of Anhui Provincial Natural Science Research Project, China (KJ2018A0198); the Foundation of Anhui Medical University, China (grant number 2019xkj134); and Basic and Clinical Cooperative Research Promotion Plan of Anhui Medical University, China (2020xkjT021). There was no additional external funding received for this study. The funders had no role in study design, data collection and analysis, decision to publish, or preparation of the manuscript.

==============================
Background

Diffuse large B-cell lymphoma (DLBCL) is a highly heterogeneous malignancy with varied outcomes. However, the fundamental mechanisms remain to be fully defined.

Aim

We aimed to identify core differentially co-expressed hub genes and perturbed pathways relevant to the pathogenesis and prognosis of DLBCL.

Methods

We retrieved the raw gene expression profile and clinical information of GSE12453 from the Gene Expression Omnibus (GEO) database. We used integrated bioinformatics analysis to identify differentially co-expressed genes. The CIBERSORT analysis was also applied to predict tumor-infiltrating immune cells (TIICs) in the GSE12453 dataset. We performed survival and ssGSEA (single-sample Gene Set Enrichment Analysis) (for TIICs) analyses and validated the hub genes using GEPIA2 and an independent GSE31312 dataset.

Results

We identified 46 differentially co-expressed hub genes in the GSE12453 dataset. Gene expression levels and survival analysis found 15 differentially co-expressed core hub genes. The core genes prognostic values and expression levels were further validated in the GEPIA2 database and GSE31312 dataset to be reliable (p < 0.01). The core genes’ main KEGG (Kyoto Encyclopedia of Genes and Genomes) pathway enrichments were Ribosome and Coronavirus disease-COVID-19. High expressions of the 15 core hub genes had prognostic value in DLBCL. The core genes showed significant predictive accuracy in distinguishing DLBCL cases from non-tumor controls, with the area under the curve (AUC) ranging from 0.992 to 1.00. Finally, CIBERSORT analysis on GSE12453 revealed immune cells, including activated memory CD4+ T cells and M0, M1, and M2-macrophages as the infiltrates in the DLBCL microenvironment.

Conclusion

Our study found differentially co-expressed core hub genes and relevant pathways involved in ribosome and COVID-19 disease that may be potential targets for prognosis and novel therapeutic intervention in DLBCL.

Introduction

Diffuse large B-cell lymphoma (DLBCL) is exceptionally heterogeneous and the most common aggressive non-Hodgkin lymphoma (NHL) subtype in adults. It is increasingly appreciated that its varied outcomes depend on the patients’ clinical and biological features (Karube et al., 2018; Liu et al., 2019b; Luo et al., 2018; Naresh et al., 2011). Despite several reports on the mechanism of DLBCL, its pathogenesis characterized by multiple abnormalities at different molecular levels remains unresolved. Its development and progression are multifaceted, comprising various signaling pathways and driver genes. Despite improved clinical outcomes with current therapies, such as rituximab-chemotherapy (R-CHOP), chimeric antigen receptor (CAR)-T cell, and advancement in stem cell transplantation, over 40% of high-risk patients relapse or develop the primary refractory disease. Mortality figures remain high (Karube et al., 2018; Luo et al., 2018). Therefore, an in-depth understanding of disease biology could reveal novel biomarkers of diagnostic and prognostic value. It will also facilitate the design of alternative personalized therapeutic strategies for DLBCL.

Advances in gene profiling technologies, high-throughput data, and bioinformatics databases make screening DLBCL for differentially co-expressed genes indispensable, particularly when integrated with personalized genomic profile data (Lui et al., 2015; van Dam et al., 2018). Recently, Liu et al. (2019a) identified eleven genes associated with endometrial cancer progression and prognosis by comprehensive bioinformatics analysis. Zhou et al. (2019) used CIBERSORT and other bioinformatics analyses for colon cancer. They found that the tumor microenvironment (TME) was abundantly enriched with M0 and M2 macrophages, activated memory CD4+ T cells, and other immune cells that could play crucial roles as biomarkers (Zhou et al., 2019). However, few integrated bioinformatics studies have compared the gene expression profile of DLBCL with non-cancer controls.

Thus, we downloaded the Gene Expression Omnibus (GEO) raw dataset of GSE12453 and compared 11 DLBCL cases with 24 non-cancer controls (non-neoplastic B lymphocytes isolated from blood or tonsils). We performed a series of screens and analyses, including filtering off differentially expressed genes (DEGs), enrichment analysis, and co-expression analysis to determine hub genes of clinical significance to DLBCL. We identified 15 differentially co-expressed core hub genes associated with the prognosis of DLBCL (RPS24, RPS21, RPL31, RPL30, RPS17, MRPS28, FAU, RPS25, RPL22L1, NDUFA6, CXCL9, CCL4, MRPL33, HEBP1, and RPL11). The KEGG (Kyoto Encyclopedia of Genes and Genomes) analysis associated most of the genes with Ribosome and Coronavirus disease-COVID-19 pathways. Validation in the GEPIA2 database and GSE31312 dataset revealed that the core genes had consistent expression levels and were reliable. Receiver operating characteristic (ROC) curves plotted demonstrated that the core genes could be potential diagnostic biomarkers. The identified genes could play critical roles in diagnosis, prognosis, and help establish a foundation for developing or identifying novel targeted therapies for DLBCL.

Materials and Methods

Data collection

We evaluated and downloaded the raw gene expression profiles from the National Center for Biotechnology Information-Gene Expression Omnibus (NCBI-GEO) database (https://www.ncbi.nlm.nih.gov/geo/). The series was based on GPL570 [HG-U133_Plus_2] Affymetrix Human Genome U133 Plus 2.0 Array. GSE12453 (Brune et al., 2008) was used to identify differentially expressed and co-expressed genes. It was also used to predict tumor-infiltrating immune cells (TIICs). It contained 11 DLBCL cases and 25 normal controls. The controls were non-neoplastic B lymphocytes isolated from healthy donors’ blood or tonsils from routine tonsillectomy patients. Similar expression profiles on the selected GPL570 platform contained normal but reactive controls or non-human controls, such as cell lines or limited controls, and were excluded. We used the GSE31312 (Visco et al., 2012) with 498 DLBCL cases for survival analysis and identifying immune cells infiltrating the TME.

Study design and data pre-processing

The GSE12453 CEL file was pre-processed using the Affymetrix package (Gautier et al., 2004) in R software (https://www.r-project.org/) version 4.0.2 (Team, 2019). The procedures included background correction, log2 transformation, followed by quantile normalization. We performed a standard quality assessment, including scaling factors and NUSE plots, and hierarchical clustering to identify outliers. The study’s design flowchart is shown in Fig S1.

Screening of differentially expressed genes (DEGs)

The DEGs between DLBCL cases and non-cancer controls were screened with R package limma (https://www.bioconductor.org/), Release 3.11 (Ritchie et al., 2015). The cutoff criteria were p < 0.05, |log2fold change (FC)| > 1.0. The pheatmap package was used to generate hierarchical clustering, and ggplot2 (Wickham, 2016) was used to show the volcano plot in R. We used the resulting data output tables that included gene ID, log2FC, unadjusted and adjusted p-values in subsequent analyses.

Functional and pathway enrichment analysis

We investigated the functional roles and pathway signaling relevance of the DEGs. The Gene Ontology (GO) and Kyoto Encyclopedia of Genes and Genomes (KEGG) pathway enrichment analysis was performed using the R clusterprofiler (https://bioconductor.org/), Release 3.12 (Yu et al., 2012). A p-value < 0.05 was considered significantly enriched. The GO categories included biological process (BP), molecular function (MF), and cellular component (CC).

Gene set enrichment analysis (GSEA)

We used the normalized expression dataset of GSE12453 for the GSEA (www.gsea-msigdb.org/gsea/index.jsp), version 4.1.0. We followed the recommended protocol. This included “gct” and “cls” file formats for the expression dataset and phenotype labels, respectively (Subramanian et al., 2005). Significant gene sets had a false discovery rate (FDR) <0.25 and a nominal p-value < 0.05.

Protein-protein interaction (PPI) network of DEGs and hub modules selection

We applied the STRING database (https://string-db.org), version 11.0 (Szklarczyk et al., 2019), to assess and map the identified DEGs into a human PPI network. Then, we uploaded the resulting data output into Cytoscape software (https://cytoscape.org/), version 3.8.2 (Shannon et al., 2003), and used MCODE with default parameters (Bader & Hogue, 2003) and the Cytohubba (Chin et al., 2014) plug-ins to select significant modules and top-ranked genes. Nodes and edges represented genes and their interactions.

Predicting tumor-infiltrating immune cells (TIICs)

We applied the Cibersort algorithm (Newman et al., 2019) in R to predict TIICs using the normalized GSE12453 dataset, according to the CIBERSORT instructions. We used data with a p-value < 0.05 for further analysis.

Co-expression network construction and identification of modules related to DLBCL

We conducted gene co-expression analysis on the processed GSE12453 data using the weighted correlation network analysis (WGCNA) (Langfelder & Horvath, 2008; Langfelder & Horvath, 2012) in R (Team, 2019). We followed the standard protocols, including quality control procedures. We performed a power β transformation on the computed Pearson correlation matrix to ensure a scale-free topology. The minimum number of module genes was set at 30. The WGCNA R package then generated a co-expression network from the resulting adjacency matrix. We applied the dynamic Tree Cut package (Team, 2019) to create the co-expression modules from the color-coded hierarchical clustering dendrogram. We assessed clinical module-trait relationships with Pearson’s correlation. Gene significance (GS) and module membership (MM) were also analyzed for their correlation in modules. Statistically significant modules were defined as p < 0.05.

Identifying differentially co-expressed hub module genes

The hub genes were identified from the intersection between the DEGs and genes significant in the WGCNA modules. We then analyzed the hub genes with the STRING and Cytoscape databases. We further conducted functional and pathway analysis on the hub genes to determine the relevant genes that impact DLBCL.

Validation and survival analysis of core hub genes

We validated the expression levels of the hub genes in the GEPIA2 (gepia2.cancer-pku.cn/#index) database (Tang et al., 2019). We retrieved the pre-processed quantile normalized series matrix file (GSE31312, Affymetrix HG-U133 Plus 2.0 GeneChips) containing 498 de-novo adults DLBCL from the NCBI-GEO database. We used it for the prognostic value analysis (overall survival (OS) and progression-free survival (PFS)). We plotted Kaplan–Meier survival curves with ggplot2 in R. The genes with p-value < 0.05 were selected as core hub genes. Also, receiver operating characteristic (ROC) curves were plotted with the pROC R package (Team, 2019) to validate the diagnostic value of the core genes. Then, the human protein atlas (proteinatlas.org/) was used to confirm their protein expressions in some selected lymph node samples (Uhlen et al., 2017).

We used the ssGSEA (single-sample Gene Set Enrichment Analysis) package (Subramanian et al., 2005) in R on the GSE31312 dataset to investigate immune-associated core hub genes in the TME.

In addition, we also employed the GEPIA2 and Oncomine (Rhodes et al., 2004) databases to determine the expression levels and relevance of the core hub genes in other tumors.

Results

Differentially expressed genes (DEGs)

According to standard protocols, one poor quality control sample (GSM312887) was excluded from the GSE12453 dataset after pre-processing. We identified 1,260 DEGs between the 11 DLBCL and 24 non-tumor controls. They comprised 1,014 up-regulated and 246 down-regulated genes (p < 0.05 and |log2FC| > 1). The gene list, Affymetrix probe ID, and log FC are shown in (Excel S1). The expression patterns of the DEGs are shown in Fig. 1. Figure 1A shows the volcano plot of all expressed genes. The DLBCL cases showed a distinctive gene expression profiling. As shown in Fig. 1B, the heatmap of the top 169 DEGs with |log2FC| > 2 suggested that the identified DEGs expression levels could differentiate DLBCL from non-tumor samples. We utilized the CytoHubba application in Cytoscape, employing five calculation methods: the Maximal Clique Centrality (MCC), Maximum Neighborhood Component (MNC), Degree, Edge Percolation Component (EPC), and EcCentricity to rank the top 250 DEGs. The genes from the five methods were intersected using the Venn diagram software (http://bioinformatics.psb.ugent.be/webtools/Venn/). Most of the intersecting genes (Fig. 1C) were associated with significant and valuable pathways such as the proteasome, spliceosome, and viral protein interaction with cytokine and cytokine receptor (Fig. S2). Intersecting genes are common genes with a high degree of interconnection and are more likely to represent key candidate genes with important biological regulatory functions.

Figure 1 Statistics for the differentially expressed genes.

(A) Volcano plot highlighting significant genes in DLBCL and non-tumor tissues. UP represents upregulated; DOWN, downregulated; NOT, not significant. (B) Heatmap of the top 169 DEGs between DLBCL cases and controls (NORMAL); |log2FC| > 2, p-value <0.05; the range of the colors corresponds with the range of expression values. (C) Venn diagram shows 118 overlapping (common) DEGs screened using five Cytohubba centrality methods. DEGs, differentially expressed genes.

Functional and pathway analysis

We applied the enrichGO or enrichKEGG function of the clusterProfiler package (Yu et al., 2012) in R to investigate the biological functions of all the DEGs (p < 0.05). The GO and KEGG pathway analysis results are shown in Fig. 2. In GO analysis, the DEGs were primarily enriched in ATP synthesis coupled electron transport and mitochondria ATP synthesis coupled electron transport for biological processes (BP). Their cellular components (CC) were mainly related to the mitochondria protein complex, mitochondria inner membrane, etc. Their molecular functions (MF) consisted of structural constituents of ribosomes and NADH dehydrogenase activity (Fig. 2A). The KEGG pathway analysis showed significant enrichment in ribosome and oxidative phosphorylation (Fig. 2B). Strikingly, the DEGs involved in the KEGG pathway and GO enrichment analysis were mainly upregulated genes, with few down-regulated genes.

Figure 2 Gene Ontology (GO) and KEGG enrichment analysis of all differentially expressed genes (DEGs).

(A) In GO analysis, the top 10 significantly enriched DEGs. The x-axis is the number of DEGs involved in the GO terms; the y-axis is the significantly enriched GO terms. (B) In KEGG analysis, the top 15 significantly enriched pathways of the DEGs. KEGG, Kyoto Encyclopedia of Genes and Genomes.

Further analysis showed that the down-regulated genes were enriched in B cell activation (adjust p-value = 0.003) for GO-BP; our data showed no other significant GO and KEGG enrichment for the down-regulated genes. The GO and KEGG enrichment analysis for the upregulated genes was similar to the analysis we performed earlier. The GO and KEGG analysis classifies genes into functional categories to help understand their functions and regulatory pathways. Hence, with additional investigations, the genes in these pathways might throw more light on the pathogenesis of DLBCL.

Gene Set Enrichment Analysis (GSEA) of DLBCL expression dataset (GSE12453)

We performed a GSEA analysis to compare the DLBCL and non-tumor controls’ expression profiles to understand better the biological functions of the relevant genes discovered. We analyzed all the qualified genes in the GSE12453 expression dataset. The KEGG output from the GSEA was similar to our previous pathway analysis and confirmed our earlier results. The Hallmark gene sets showed immune and metabolic-related signaling predominance, including estrogen response late, epithelial-mesenchymal transition, and UV response up (Table S1). Interestingly, genes defining late response to estrogen, epithelial-mesenchymal transition (such as in fibrosis and metastasis), and genes upregulated in response to ultraviolet (UV) radiation have not been fully elucidated in DLBCL. However, evidence suggests these genes have essential roles in oncogenesis. These findings provide evidence that can drive future research with therapeutic implications.

Predicting tumor-infiltrating immune cells (TIICs)

Our GO, KEGG, and GSEA analyses showed that the DEGs were enriched in some immune-related biological functions. Immune cell infiltration into tumors plays an essential role in tumorigenesis and metastasis. So, we applied the CIBERSORT algorithm to the GSE12453 dataset to predict immune cells infiltrating the TME. Among the immune subsets analyzed, activated memory CD4+ T cells, CD8+ T cells, and M0, M1, and M2 macrophages were the most represented cell fractions within the DLBCL microenvironment (Figs. 3A, 3B). The correlations among the TIICs ranged from high to negligible (Fig. S3). Regulatory T cells (Tregs) showed a moderate negative correlation with activated memory CD4+ T cells. However, M0 macrophages had a high positive correlation with activated memory CD4+ T cells and M1 macrophages. The varied infiltrating immune cell types could be reflective of the complexity and the unusual behavior of DLBCL. Uncommitted macrophages (M0) can polarize into the M1 (considered tumoricidal) and M2 (pro-tumorigenic) to show paradox effects on tumor prognosis (Dancsok et al., 2020). Memory B cells and activated dendritic cells were the most represented fractions in the non-tumor controls (Fig. 3A). The findings suggest that TIICs may be closely associated with clinical outcomes. Future studies, including their correlation to DLBCL disease stages, will be meaningful, particularly for immunotherapy.

Figure 3 The prediction of tumor-infiltrating immune cells (TIICs) using the GSE12453 dataset.

Violin plot comparing the proportions of TIICs between non-tumor controls (in blue ) and DLBCL (in red). The x and y axes represent TIICs and their relative percentages, respectively. There was no T cell CD4 memory resting. (B) Bar plots for 24 non-tumor controls and 11 DLBCL samples (x-axis) and the percentages of immune cell subsets (y-axis).

Protein-protein interaction (PPI) network and hub modules establishment

We constructed the PPI network of all DEGs using the STRING database’s multiple proteins function to determine genes likely to perform biological functions together. The highest confidence of 0.9 was set with unconnected nodes taken out. As shown in Fig. 4A, it yielded 609 nodes and 8,583 edges. These genes were highly inter-connected than expected (PPI enrichment p-value < 1.0e−16). We observed six important clusters with a k-score > 10 when we analyzed the tab-separated values (tsv) file with Cytoscape’s MCODE. The largest cluster (#1) had the highest score of 33.17 (Fig. 4B); it mainly comprised Ribosome genes (FDR = 3.49e−83) in KEGG analysis. Cluster 2 (Fig. 4C) was enriched in genes associated with oxidative phosphorylation (FDR, 7.23e−56) and Parkinson’s disease (FDR, 2.62e−55). Cluster three (Fig. 4D) genes were mainly involved in chemokine signaling (FDR, 1.06e−22). We identified 109 highly connected DEGs (Excel S1) from these top three clusters. These DEGs could play essential roles in DLBCL, so they were selected for further hub gene screening.

Figure 4 The protein-protein interaction (PPI) networks using the STRING and Cytoscape databases.

(A) The PPI network for the 109 highly connected DEGs; confidence score, 0.9. (B) Cluster one consisted of 61 nodes, 995 edges with the highest k-score of 33.17. (C) Cluster two had 28 nodes, 359 edges, and a k-score of 26.59. (D) Cluster three had 20 nodes, 190 edges, and a k-score of 20.00. (E) The top 250 ranked DEGs.

In this study, the down-regulated DEGs were not part of the constructed PPI, so we determined their relevance in DLBCL. We ranked the top 60 DEGs (30 up-and 30 down-regulated) by log2FC and analyzed them in the STRING database (Fig. S4A). The only down-regulated DEG in the network built was identified upregulated when verified in the GEPIA2 database. To further investigate the down-regulated DEGs’ functional relatedness, a PPI was constructed for all the down-regulated DEGs (Fig. S4B). We found that the down-regulated genes were enriched in B cell activation for GO-BP, shown in (Fig. S4C). Finally, we used the degree method of the CytoHubba application to predict the top 250 important genes (Fig. 4E). Genes with a high degree of centrality are vital since they have many direct interacting gene partners. If confirmed, these critical findings could improve the general understanding and the potential causes of variation in the clinical prognosis of DLBCL.

Weighted gene co-expression network (WGCNA) analysis

To identify co-expression modules that could share similar biological functions or regulatory mechanisms with clinical relevance to DLBCL, we applied the WGCNA package (Langfelder & Horvath, 2008; Langfelder & Horvath, 2012) in R (Team, 2019). The GSE12453 dataset we processed was used. We carried out quality control procedures, including inspecting good genes and sample hierarchical clustering to detect potential outliers, but no obvious outliers were found (Fig. 5A). The 35 samples yielded two main clusters. We applied the WGCNA on the top 25% of the 21,654 expressed genes ranked by the largest variance. To satisfy a scale-free network topology, we choose the soft-threshold power β of eight with R2 = 0.86 (Figs. 5B, 5C). Hierarchical clustering and the dynamic tree-cutting yielded 18 modules of co-expressed genes (Fig. 5D). Finally, we visualized the top 1,000 significantly expressed genes with a heatmap (Fig. 5E); they represent interesting genes for further analysis.

Figure 5 Construction of gene co-expression network.

(A) Sample clustering to detect outliers, no obvious outliers found. (B and C) Determination of soft-threshold power. When β is set at eight, the log-log plot of the network connectivity distribution produces a straight line. (D) Hierarchical clustering dendrograms (top modules). Each color band (bottom) represents a color-coded module that contains a group of highly connected genes. The Dynamic Tree Cut identified 18 modules. (E) A heatmap showing the topological overlap matrix (TOM) among the top 1,000 genes selected from all genes. The color intensity indicates the correlation strength between pairs of modules: the left side (gene dendrogram) and the top side (module assignment).

To investigate the molecular mechanisms of the traits, we correlated each Module Eigengenes (ME) to disease status (DLBCL and non-tumor controls). The results are shown in Fig. 6. The ME turquoise and green (Figs. 6A, 6B) containing 292 and 72 genes, respectively, strongly correlated with DLBCL. ME dark magenta with five genes had the strongest negative correlation. The cut-off was set at gene significance (GS) value >0.8, and absolute Module Membership (MM) value >0.7. Besides, the GS versus MM plots for these three modules were highly correlated (Fig. 6C), reflecting their high association with DLBCL. We selected these three clinically significant modules with the 369 high connectivity genes (gene list shown in Excel S1) for further analysis. The highly connected genes are often the most important (central) elements of the respective modules and tend to play key roles in the biological processes. The ME genes are listed in Excel S1. Altogether, these co-expressed genes might provide new clues to understand the biology of DLBCL in the future.

Figure 6 Identifying modules of clinical relevance from the GSE12453 dataset.

(A) Module trait relationship showing correlation coefficients between module eigengenes (row) and disease status (column), with the corresponding p-values in brackets. The degree of correlation is based on a color legend: red, strong positive and blue, strong negative correlation. (B) Heatmap plot of the adjacencies in the eigengene network, including the relationship with DLBCL trait. The top panel is the hierarchical clustering dendrogram of the eigengenes. The bottom panel shows the eigengene adjacency. (C) Scatter plots of gene significance (GS) versus module membership (MM) for the DLBCL related modules (turquoise, green and dark magenta, respectively).

PPI and functional enrichment analysis of the WGCNA relevant modules

The 369 high connectivity genes from the three relevant modules were filtered in the STRING database followed by the Cytoscape; the network yielded a PPI with 195 nodes and 1,579 edges. The PPI and gene list are detailed in Fig. S5 and Excel S1. These 195 genes were considered functionally important. As presented in (Fig. 7), functional annotation revealed that these genes were involved in viral transcription and viral gene expression in the BP category. In KEGG analysis, the genes were primarily enriched in ribosome, coronavirus disease-COVID-19, and oxidative phosphorylation. The identified pathways were roughly consistent with that of the DEGs. These processes and signaling pathways are usually disrupted in cancer and could provide an insight into the pathogenesis of DLBCL.

Figure 7 Top 10 enrichment results of the 195 WGCNA (weighted correlation network analysis) genes.

(A) Gene Ontology (GO) functional analysis. (B) Kyoto Encyclopedia of Genes and Genomes (KEGG) pathway analysis.

Identification of hub genes and pathways

Eventually, we identified 46 important differentially co-expressed genes by the Venn diagram software, as shown in (Fig. 8A). These 46 genes were common between the DEGs and WGCNA hub module genes and were regarded as hub genes. We re-analyzed the 46 genes with the STRING and Cytoscape databases, and the PPI is shown in (Fig. 8B). Their GO and KEGG enrichments in the R software (Table 1) were similar to the other analyses. The KEGG common genes in the ribosome, COVID-19, and oxidative phosphorylation pathways are shown in Table 2. The above pathway genes play essential roles in metabolic reprogramming and tumor-promoting inflammation of cancer and warrant further studies.

Figure 8 Differentially co-expressed hub genes.

(A) Venn diagram indicating the 46 common genes. (B) Protein-protein interaction (PPI) network of the 46 hub genes (STRING and Cytoscape databases).

Table 1 Gene Ontology and KEGG pathway analysis of the 46 differentially co-expressed genes.

Term	Description	Count	p. adjust	
Biological process			
GO:0006614	SRP-dependent cotranslational protein targeting to membrane	16	1.29E−22	
GO:0019083	Viral transcription	18	1.29E−22	
GO:0006613	Cotranslational protein targeting to membrane	16	1.29E−22	
GO:0019080	Viral gene expression	18	2.85E−22	
GO:0045047	Protein targeting to ER	16	2.98E−22	
GO:0000184	Nuclear-transcribed mRNA catabolic process, nonsense-mediated decay	16	3.30E−22	
GO:0072599	Establishment of protein localization to endoplasmic reticulum	16	3.74E−22	
GO:0070972	Protein localization to endoplasmic reticulum	16	7.45E−21	
Cellular component			
GO:0044391	Ribosomal subunit	25	2.39E−37	
GO:0005840	Ribosome	25	1.45E−33	
GO:0022626	Cytosolic ribosome	18	3.96E−28	
GO:0015934	Large ribosomal subunit	17	1.54E−25	
GO:0044445	Cytosolic part	18	6.76E−22	
GO:0098798	Mitochondrial protein complex	17	8.16E−20	
GO:0005743	Mitochondrial inner membrane	19	2.01E−18	
GO:0022625	Cytosolic large ribosomal subunit	11	1.14E−17	
Molecular function			
GO:0003735	Structural constituent of ribosome	24	1.15E−33	
GO:0003954	NADH dehydrogenase activity	9	3.12E−14	
GO:0008137	NADH dehydrogenase (ubiquinone) activity	9	3.12E−14	
GO:0050136	NADH dehydrogenase (quinone) activity	9	3.12E−14	
GO:0016655	Oxidoreductase activity, acting on NAD(P)H, quinone or similar compound as acceptor	9	3.26E−13	
GO:0016651	Oxidoreductase activity, acting on NAD(P)H	9	6.05E−11	
GO:0008009	Chemokine activity	6	2.25E−08	
KEGG pathway			
hsa03010	Ribosome	24	4.36E−29	
hsa05171	Coronavirus disease-COVID-19	21	2.35E−20	
hsa00190	Oxidative phosphorylation	11	9.15E−10	
Note:

The selection of terms enriched in the categories was based on the most significant adjusted p-value (p. adjust). Count: the number of genes enriched in each term or pathway.

Table 2 List of the hub genes involved in the three KEGG pathways.

S/N	24 differentially co-expressed (DCE) ribosome genes	21 DCE coronavirus disease-COVID-19	11 differentially co-expressed oxidative phosphorylation genes	
1	MRPS18A	RPL10L	UQCRQ	
2	RPL10L	RPL24	NDUFB1	
3	MRPL17	RPS17	NDUFB2	
4	RPL24	CXCL10	NDUFB3	
5	RPS17	RPL30	NDUFS6	
6	RPL30	RPS18	NDUFA1	
7	RPS18	C3	NDUFA3	
8	RPS19	RPS19	NDUFA11	
9	RPL29	RPL29	NDUFB9	
10	RPL31	RPL31	NDUFA6	
11	RPL36	RPL36	UQCR11	
12	MRPL33	RPS21		
13	RPS21	FAU		
14	MRPL21	RPS24		
15	FAU	RPL14		
16	RPS24	RPS25		
17	RPL14	RPL22L1		
18	RPS25	RPL11		
19	RPL22L1	RPL35		
20	RPL35	RPL12		
21	MRPL14	RPS7		
22	RPL11			
23	RPL12			
24	RPS7			
Note:

DCE, differentially co-expressed.

Validation of expressions and prognostic analysis of core hub genes

We applied GEPIA2 to validate the reliability and authenticity of the 46 hub genes in the cancer genome atlas (TCGA) dataset. We identified 44 prognostic genes with higher expression (consistent with that in the GSE12453 dataset) in DLBCL tissues than the non-tumor control tissues (p < 0.01) (Fig. 9) (15 genes shown) and (Excel S1). Kaplan–Meier survival analysis on the GSE31312 showed 15 of these genes (p < 0.05) correlated with patient outcomes (Fig. 10 & Table S2). Except for RPL11, the patients with high expressions had significantly shorter 5-year OS and PFS, suggesting these genes are potential oncogenes and have a role in DLBCL development and/or progression. The 15 genes (Fig. 11A) were considered as the core hub genes. Moreover, ROC curve analysis for their diagnostic potentials obtained AUCs ranging from 0.992 to 1.00, indicating optimal performance to accurately differentiate DLBCL from non-tumor control cases (Fig. S6). Also, immunohistochemistry data from the human protein atlas (HPA) database demonstrated the protein expressions of some of the genes in some lymph node samples with cytoplasmic/membranous localization (Fig. S7, four genes shown); data were retrieved from https://www.proteinatlas. The genes included RPS21, MRPS28, RPL31, and RPL30. As expected, they would be involved in metabolic pathways such as glycolysis and processes including signal transduction and cell division.

Figure 9 The expressions of the hub genes in the GEPIA2 database.

*(p < 0.01). The data were retrieved from the GEPIA2 database (http://gepia2.cancer-pku.cn/#index).

Figure 10 The Kaplan–Meier estimates for the overall survival (OS) of the 15 core hub genes in GSE31312 (p < 0.05).

Figure 11 PPI and enrichment analysis of the 15 core hub genes.

(A) Cluster analysis. Genes in the circle represent covid 19 pathway genes. (B) The top five Gene Ontology (GO) terms. (C) The top three KEGG (Kyoto Encyclopedia of Genes and Genomes) pathways.

The core hub genes’ functional annotation was mainly associated with Ribosome and Coronavirus disease-COVID-19 (Figs. 11B, 11C; Table 3). To assess the tumorigenic potentials of the COVID-19 genes regarding immune cells infiltrating the TME, the ssGSEA analysis was applied (GSE31312). As shown in Fig. S8, seven out of the nine COVID-19 pathway genes negatively correlated with mast cells, five with immature dendritic cell (iDC), and three genes negatively correlated with plasmacytoid DC. RPL30, RPL31, RPS25, and FAU positively correlated with tumor-infiltrating lymphocytes (TIL) and macrophages. RPL30, RPL31 correlated with Tregs. These infiltrating immune cells may be involved in regulating tumor proliferation, dormancy, and drug resistance.

Table 3 List of the core hub genes involved in the three KEGG pathways.

KEGG pathway	Number of genes	List of genes	p. adjust	
hsa03010: Ribosome	10	RPS24/RPS21/RPL31/RPL30/RPS17/FAU/RPS25/RPL22L1/MRPL33/RPL11	3.12E-14	
hsa05171: Coronavirus disease-COVID-19	9	RPS24/RPS21/RPL31/RPL30/RPS17/FAU/RPS25/RPL22L1/RPL11	6.89E-11	
hsa04061: Viral protein interaction with cytokine and cytokine receptor	2	CXCL9/CCL4	0.0552	

Finally, we determined whether the core genes were upregulated in other tumors. The GEPIA2 database revealed that all the 15 core genes were upregulated in thymoma (THYM), and 11 genes were upregulated in testicular germ cell tumors (TGCT). Notably, RPL30 and FAU genes were consistently upregulated in all six different cancer types identified (Table S3). In the Oncomine database, some of the core genes were upregulated in various lymphoma datasets and other cancers, including Sarcoma (Fig. S9). The results suggest that the upregulation of these 15 hub genes may not be limited to DLBCL.

Discussion

Diffuse large B-cell lymphoma (DLBCL) remains a significant clinical challenge; over 30% of patients are not cured (Pasqualucci & Dalla-Favera, 2018; Yi et al., 2020). So far, no functional assays capable of screening exit, so effective management is required once diagnosed. Hence, identifying unique gene signatures and regulatory pathways related to its pathogenesis and prognosis is meaningful. Here, we examined the gene expression profile of GSE12453 to find dysregulated common core hub genes and pathways to help further understand DLBCL pathogenesis and provide potential biomarkers.

Integrated bioinformatics analysis, gene expression levels, and survival analysis identified 15 differentially co-expressed core hub genes linked to DLBCL pathogenesis. The genes included RPS24, RPS21, RPL31, RPL30, RPS17, MRPS28, FAU, RPS25, RPL22L1, NDUFA6, CXCL9, CCL4, MRPL33, HEBP1, and RPL11. Their primary KEGG enrichment was ribosome and coronavirus disease-COVID-19, which was in line with the other analyses. The construction of ROC curves yielded very high AUC values suggesting the genes could accurately distinguish between DLBCL and non-tumor control cases and might be potential biomarkers. In addition, experimentally derived data from the HPA by IHC indicated the protein expression of some of the genes in some lymph node samples. RPS21, MRPS28, RPL31, and RPL30 showed relatively higher protein expressions in some DLBCL and other malignant lymphoma tissues than the averaged expressions in normal tissues, though not significant. The HPA database sample size was limited; however, the HPA experimental findings can be extended to DLBCL and other lymphomas, thus providing a valuable basis for medical and biological research.

Lastly, most of the core genes were upregulated in different cancer types. Cancer is a complex disease, so the genes might have similar or different prognostic roles in these tumors. The overall survival data from GEPIA2 demonstrated that low levels of FAU, RPS17, and RPS 24 were significantly associated with shorter survival, while high CCL4 was significantly associated with shorter survival in thymoma patients. Nonetheless, the genes’ potential biological and clinical relevance is not restricted to only DLBCL. These genes could be prognostic markers and therapeutic targets across different tumor types, particularly for patients with multiple coexisting tumors.

Little is known experimentally about the roles of most of the core genes proteins in DLBCL. However, dysregulated ribosomal proteins have been reported to play various critical roles in other tumors (Wang et al., 2015). Among our ribosome genes, over-expressed RPS21 promoted prostate cancer (PCa) cell proliferation, migration, and invasion, inhibited PCa cell apoptosis, and was suggested as a promising biomarker, with a potential application in diagnosis or treatment (Liang et al., 2019). The 8q-mapped RPL30 gene was associated with adverse outcomes in Medulloblastoma patients (De Bortoli et al., 2006). RPS24 significantly promoted colorectal cancer (CRC) cells’ proliferation rate and increased CRC risk in patients (Zou et al., 2020). The knockdown of RPS24 inhibited cell proliferation and cell migration in human CRC cell lines and was recommended as a biomarker (Wang et al., 2015). A study also implicated MRPS28 in the molecular pathogenesis of bladder cancer (Liu et al., 2021).

The coronavirus disease-COVID-19 and viral transcription enrichments agree with recent studies implicating various viruses (Gandhi et al., 2020) (Fedoriw et al., 2020) in the development and progression of DLBCL subtypes. Most DLBCL patients have an underlying immune dysfunction and can easily get viral infections. Viruses such as COVID 19 could manipulate the function of the COVID 19 related genes in the TME. Besides, the Gene Cards database (https://www.genecards.org/) demonstrated all the covid 19 pathway genes (RPS24, RPS21, RPL31, RPL30, RPS17, FAU, RPS25, RPL22L1, RPL11) are related to viral mRNA translation (Stelzer et al., 2016). Additionally, the DAVID database (Protein interactions) associated five COVID 19 genes (RPL30, RPS17, RPS24, RPS25, and FAU) with HIV interactions (Huang, Sherman & Lempicki, 2009).

Recently, hematological malignancies (HM) patients were reported to have a more severe COVID-19 trajectory than patients with solid organ tumors (Lee et al., 2020). A significant number of the COVID 19 related genes were upregulated in various lymphomas and some multiple myeloma datasets (Fig. S9). Most of our COVID-19 pathway genes showed some correlations with immune infiltrates such as TIL and macrophages. An anti-viral immune response can have protective effects with improved survival in coronavirus infection, but excessive inflammation can be harmful (“cytokine storm”). High pro-inflammatory macrophage (M1) and low CD8+ T cells were observed in the microenvironment of severe/critical COVID-19 patients (Liao et al., 2020). Higher expression of CXCL9 in COVID-19 patients than healthy controls and higher levels of CCL4 in severe COVID-19 patients were also found (Liao et al., 2020). These are partly consistent with our data and previous knowledge on various cancers (Brune et al., 2008; Chang et al., 2013; De la Fuente López et al., 2018). Not much is known about COVID-19 and most cancers, including DLBCL pathogenesis. However, it is tempting to speculate that COVID-19 infection, together with the COVID-19 related genes, could increase macrophage polarization to M1 (hyper-inflammatory response) to worsen prognosis (He et al., 2020; Passamonti et al., 2020; Shah et al., 2020). But, the mechanism is unclear, and limited data on the topic did not permit detailed discussion. However, extensive genome-related studies are required to verify this association between COVID 19 and DLBCL; these genes could provide a basis to identify effective preventive and therapeutic strategies.

Clinical implication analysis in the GEPIA2 database showed that the core hub genes were significantly overexpressed in DLBCL. The high expressions of 14 (93%) were negatively associated with prognostic outcomes (worse OS and PFS times). These emphasize their potential role as oncogenes and could be utilized as prognostic indicators for DLBCL. The high expression of RPL11 might be associated with a favorable clinical outcome (Kawahata et al., 2020; Kayama et al., 2017). They offer unique opportunities for further investigation.

In addition to the ribosome (translation), the 46 hub genes were significantly represented in oxidative phosphorylation (OxPhos) and mitochondria inner membrane. Growing evidence suggests cancer is primarily a mitochondrial metabolic disease that exhibits altered energy production and dysregulated metabolic crosstalk (Yin et al., 2019; Norberg et al., 2017). Their inhibition has demonstrated anti-cancer efficacy (Martínez-Reyes et al., 2020; Norberg et al., 2017). Thus, with further studies, these metabolic genes could be rational targets, especially for the metabolically coupled and OxPhos-DLBCL subsets, and help understand the metabolic differences in DLBCL.

The few bioinformatics analyses on DLBCL focused on the subtypes (Huang, Liu & Shen, 2019; Zhou et al., 2020) or clinical features (Xiao, Wang & Bai, 2020). However, some dysregulated genes specific to DLBCL versus non-tumor controls cannot distinguish the subtypes (Huang, Liu & Shen, 2019). Moreover, most related studies that focused on DLBCL and non-cancer controls were based on DEGs (Huang, Liu & Shen, 2019; Luo et al., 2018) and discovered entirely different core hub genes. However, the complexity of DLBCL and the emergence of novel targeted therapies warrants more predictive personalized biomarkers for precision medicine. To our knowledge, no integrated bioinformatics analysis on DLBCL and non-tumor controls has so far been reported on the common core hub genes found and immune cells associations. Thus, our finding is novel.

One potential limitation of this study is the lack of experimental validation. However, this analysis provides a theoretical basis for our future work, which will focus on experimental verification. Second, an individual study with limited DLBCL cases was used, but the hub genes, pathways, and immune cells infiltrate identified are relevant to the pathogenesis of DLBCL and cannot be ignored. However, our results should be interpreted with caution.

Conclusions

We used the integrated bioinformatics method to highlight the critical roles of differentially co-expressed core hub genes and relevant pathways in DLBCL. We identified some immune-related core hub genes linked to DLBCL pathogenesis. The core genes’ main KEGG pathway enrichments were Ribosome and Coronavirus disease-COVID-19. Their verification in GEPIA2 showed they were reliable. Nevertheless, most of the core genes were upregulated in different cancer types and hold potential biological and clinical relevance in cancers. Thus, the identified genes could be potential targets for prognosis and therapeutic intervention in DLBCL and may provide insight into the pathogenetic mechanisms in DLBCL.

Supplemental Information

Supplemental Information 1 Flow diagram of the study (GSE12453). One poor-quality sample (GSM312887) was excluded.

PPI: protein-protein interaction; DEGs: differentially expressed genes; CEGs: co-expressed genes; HPA: the human protein atlas.

Click here for additional data file.

Supplemental Information 2 Gene ontology (GO) and KEGG enrichment analysis for 118 intersecting DEGs.

(A) GO analysis; the top 10 significantly enriched GO terms of intersecting DEGs, p < 0.05. (B) The top 12 significantly enriched KEGG pathways of intersecting DEGs, p < 0.05. The horizontal axis is the number of intersecting DEGs involved in the terms or pathways; the vertical axis shows the terms or pathway names. KEGG (Kyoto encyclopedia of genes and genomes).

Click here for additional data file.

Supplemental Information 3 Correlation matrix of the analyzed tumor-infiltrating immune cells in a color legend, GSE12453.

x-and y-axes both represent tumor-infiltrating immune cells. Color legend: red (positive) and blue (negative) correlations.

Click here for additional data file.

Supplemental Information 4 Protein-protein interaction (PPI) network of downregulated genes.

(A) PPI for the top 30 up-and 30 down-regulated DEGs ranked by highest log2FC at the highest confidence of 0.9. (B) PPI for all downregulated genes at high confidence of 0.7 with disconnected nodes in the network hidden (it yielded 30 nodes, 20 edges). (C) PPI of the downregulated DEGs enriched in B cell activation for GO biological processes (medium confidence, 0.4); false discovery rate, 6.29e−15).

Click here for additional data file.

Supplemental Information 5 Protein-protein interaction (PPI) network for the 3 relevant co-expressed module genes.

The PPI network of turquoise, green and dark magenta module genes at high confidence of 0.9 with disconnected nodes hidden.

Click here for additional data file.

Supplemental Information 6 Receiver operating characteristic (ROC) curve of core hub genes.

MRPS28, RPL22L1, RPL30, and RPS17 each had an AUC of 0.989. Also, CXCL9, HEBP1, RPL11, RPS21 and RPS 24 each had an AUC of 1.00. The area under curve (AUC).

Click here for additional data file.

Supplemental Information 7 Immunohistochemistry (IHC) showing lymph node expression patterns of 4 core hub genes.

The averaged protein expression levels of normal tissue (NT)-not otherwise stated (NOS) non-germinal center cells (N-GCC) were relatively lower than the expression levels in some selected tumor tissues. Q: quantity of IHC score; DLBCL: NHL high grade; LG: NHL-low grade; HD-NOS: Hodgkin’s disease-NOS; id: Patient id. Data retrieved from the human protein atlas (https://www.proteinatlas.org/).

Click here for additional data file.

Supplemental Information 8 Correlation matrix predicting core hub genes associated with tumor-infiltrating immune cells in GSE31312 dataset.

Click here for additional data file.

Supplemental Information 9 The 15 core hub genes expression levels in other tumors.

Most of the genes were upregulated in various cancer types. However, some genes were downregulated in some tumors, suggesting that the same gene may have different functions in different cancers. Data were retrieved from the Oncomine database (https://www.oncomine.org/resource/main.html).

Click here for additional data file.

Supplemental Information 10 GSEA Hallmarks enrichment analysis for the GSE12453 dataset.

Top 20 hallmark enrichment. ES, Enrichment Score; NES, normalized Enrichment Score; NOM P-val, nominal p value; FDR, false discovery rate >0.25; FWER, familywise-error rate

Click here for additional data file.

Supplemental Information 11 Kaplan-Meier survival analysis of core hub genes.

Click here for additional data file.

Supplemental Information 12 List of core hub genes upregulated in other tumors.

p-value < 0.01; |Log2FC| > 1. RPL30 and FAU (marked in red) were consistently upregulated in all six tumor types. THYM: Thymoma; TGCT: Testicular Germ Cell Tumors; CHOL: Cholangio carcinoma; LGG: Brain Lower Grade Glioma; GBM: Glioblastoma multiforme; PAAD: Pancreatic adenocarcinoma (http://gepia2.cancer-pku.cn/#analysis)

Click here for additional data file.

Supplemental Information 13 Gene lists.

Click here for additional data file.

Additional Information and Declarations

Competing Interests

Author Contributions

Data Availability

The authors declare that they have no competing interests.

Alice Charwudzi conceived and designed the experiments, performed the experiments, analyzed the data, prepared figures and/or tables, authored or reviewed drafts of the paper, and approved the final draft.

Ye Meng conceived and designed the experiments, performed the experiments, analyzed the data, prepared figures and/or tables, authored or reviewed drafts of the paper, and approved the final draft.

Linhui Hu performed the experiments, prepared figures and/or tables, authored or reviewed drafts of the paper, and approved the final draft.

Chen Ding performed the experiments, authored or reviewed drafts of the paper, and approved the final draft.

Lianfang Pu analyzed the data, authored or reviewed drafts of the paper, and approved the final draft.

Qian Li analyzed the data, authored or reviewed drafts of the paper, and approved the final draft.

Mengling Xu analyzed the data, authored or reviewed drafts of the paper, and approved the final draft.

Zhimin Zhai conceived and designed the experiments, authored or reviewed drafts of the paper, and approved the final draft.

Shudao Xiong conceived and designed the experiments, authored or reviewed drafts of the paper, and approved the final draft.

The following information was supplied regarding data availability:

Previously reported expression datasets were used in this study and are available at NCBI-GEO: Dataset for differentially co-expressed genes identification: GSE12453.

Dataset for survival and prognostic verification: GSE31312.

The other databases used:

Verification of hub gene expression levels and core hub gene expression levels in other tumors; GEPIA2 database: http://gepia.cancer-pku.cn/detail.php?gene=&clicktag=expdiy [Filters used: Expression DIY; Gene (names of the various hub genes); Datasets Selection (Cancer name, e.g., DLBC); Plot]. A full list of the hub genes and cancer names is available in the Supplemental File.

Core hub gene expression levels in other tumors; Oncomine database: https://www.oncomine.org/resource/main.html (Filters used: Differential Analysis and Gene Summary. Search terms were the names of the various core hub genes).

Protein expression levels by immunohistochemistry (IHC): The human protein atlas database; https://www.proteinatlas.org/ [(A) Search terms for cases: The Pathology Atlas; names of the core hub genes (RPS21, MRPS28, RPL31, and RPL30); Pathology; Cancer: Lymphoma. (B) Search terms for normal tissues: Tissue atlas; names of the core hub genes (RPS21, MRPS28, RPL31, and RPL30); Tissues: lymph node].

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
