# Peer review of "Integrated bioinformatics analysis reveals dynamic candidate genes and signaling pathways involved in the progression and prognosis of diffuse large B-cell lymphoma"

_PeerJ, doi:10.7717/peerj.12394_

## Round 0.1 · original submission · Major Revisions

Based on the reviewers' comments, your manuscript cannot be accepted for publishing in the current status. Please revise the manuscript to address the comments from the reviewers and resubmit again.

Reviewer 1 ·

Basic reporting

Manuscript #62901 performed bioinformatic analysis on the gene expression profiles of diffuse large B-cell lymphoma (DLBCL) patients from the Gene Expression Omnibus database. By comparing to non-tumor controls, the authors claimed that they identified 15 differentially co-expressed core hub genes in DLBCL. Such analysis might be very helpful for understanding the mechanisms, guiding the diagnosis and therapy of related diseases.
The language needs to be improved. There are a lot of grammar errors, and some phrasings make comprehension difficult. For example, lines 213-215 and 227 are difficult to understand. The names of different types of T cells are normally written as CD4+ memory T cells, CD8+ T cells, and regulatory T cells, instead of the expressions in lines 235-246.
Labels in the figures are not clear, especially Figure 1, 2, 5, 6, 8, 11, 12. It is almost impossible to read the labels in these figures and get any detailed information. The descriptions in the main text are also not clear enough. For example, the authors explained the method they used for figure 1C but did not explain what information the readers can get from this figure. There are similar problems in the other figures too.

Experimental design

The experimental design and descriptions are clear. The authors used many different methods to do the analysis and presented large amounts of data in the manuscript. It might be necessary to compare the data with other tumors to see if these 15 hub genes are specifically upregulated in DLBCL or are generally upregulated in different tumors.

Validity of the findings

The authors need to include some other types of tumors as controls to see if their findings are unique to DLBCL. Some of the figures seem not relevant to the main conclusions, for example, the immunohistochemistry data in figure 13 showed the localization of the proteins in the tissues, however, I don't think it is relevant to the claims made by the authors. In addition, the images are not clear and it is hard to tell the differences among different samples.

Annotated reviews are not available for download in order to protect the identity of reviewers who chose to remain anonymous.

Reviewer 2 ·

Basic reporting

Overall, the manuscript is well written, provides the necessary background information.

Experimental design

This paper performed bioinformatics analyzes, trying to reveal some potential diagnosis, prognosis markers and novel therapeutic targets of diffuse large B-cell lymphoma (DLBCL). The authors first retrieved the raw gene expression profile from the public database, identified the differential gene expression (DGE) between the DLBCL patients and non-cancer controls, and analyzed the function and pathway. They then analyzed the infiltrating immune cell types in the tumor microenvironments and found that the T cells CD4 memory activated, T cells CD8, and macrophages were the most represented cell fractions within the DLBCL microenvironment. They further constructed the protein-protein interaction (PPI) network and finally identified 15 core hub genes, which were associated with COVID-19.

Validity of the findings

However, there are some major concerns need to be addressed.

1. A major concern, however, is about the novelty and biological significance of the results. Most of the findings were revealed previously, for example, the upregulation of p53 and KRAS genes, the immune cells infiltration profile in DLBCL patients. The paper should focus on a discussion about their novel findings and the potential biological significance of those genes.
2. The relationship between the 15 core hub genes and COVID-19 seems far-fetched based on the limited discussion provided in the manuscript. The authors should provide more convincing evidence/discussions regarding this conclusion.


Minor comments:

1. Figures are too blurry to see the details. High resolution figures should be provided.
2. The core hub genes and other major findings should be highlighted in the illustrations, so that they can stand out from thousands of genes clearly and the readers can notice them easily.

---

## Round 0.2 · Minor Revisions

The manuscript has been substantially improved. However, one of the reviewers still have concerns on some minor grammar errors and typos. Please update your manuscript to improve your writing.

Reviewer 1 ·

Basic reporting

Manuscript #62901 performed bioinformatic analysis on the gene expression profiles of diffuse large B-cell lymphoma (DLBCL) patients from the Gene Expression Omnibus database to identify differentially expressed genes in DLBCL. The revised manuscript is overall well written, provided a good background introduction. The language has been greatly improved. However, there are still some minor grammar errors and typos. Please proofread the manuscript carefully.

Experimental design

The experimental design and descriptions are clear. The authors used different methods and analyzed the data from multiple aspects. They identified the differentially expressed genes, analyzed the protein-protein network, and also looked into the tumor-infiltrating immune cell profiles in DLBCL.

Validity of the findings

The authors used many different methods to do the analysis and presented large amounts of data in the manuscript. The results are overall well explained and conclusions are well stated.

Reviewer 2 ·

Basic reporting

While I still have some reservations about the phrasing and explanation of some of the results, the overall conclusions are clear and acceptable.

Experimental design

No comment

Validity of the findings

The revisions have addressed my major questions regarding the novelty of this MS, and the core genes have been highlighted clearly.

Additional comments

The manuscript is now suitable for publication.

---

## Round 0.3 · accepted · Accept

Thanks for improving the manuscript as the reviewer suggested. I believe it is now ready for publication.